# Hyperuricemia as a Risk Factor in Hypertension among Patients with Very High Cardiovascular Risk

**DOI:** 10.3390/healthcare11172460

**Published:** 2023-09-03

**Authors:** Paweł Muszyński, Emil Julian Dąbrowski, Marta Pasławska, Marta Niwińska, Anna Kurasz, Michał Święczkowski, Justyna Tokarewicz, Łukasz Kuźma, Marcin Kożuch, Sławomir Dobrzycki

**Affiliations:** 1Department of Invasive Cardiology, Medical University of Białystok, M. Skłodowskiej-Curie 24A, 15-276 Białystok, Poland; pawel.muszynski@sd.umb.edu.pl (P.M.); anna.kurasz@sd.umb.edu.pl (A.K.); michal.swieczkowski@sd.umb.edu.pl (M.Ś.); justyna.tokarewicz@gmail.com (J.T.); lukasz.kuzma@umb.edu.pl (Ł.K.); marcin.kozuch@umb.edu.pl (M.K.);; 2Department of General and Experimental Pathology, Medical University of Białystok, Mickiewicza 2C, 15-230 Białystok, Poland; 3Department of Pediatrics, Endocrinology, Diabetology with Cardiology Divisions, Medical University of Białystok, J. Waszyngtona 17, 15-274 Białystok, Poland; marta.paslawska@sd.umb.edu.pl; 4Department of Family Medicine, Medical University of Białystok, Mieszka I 4B, 15-054 Białystok, Poland; marta.niwinska@sd.umb.edu.pl

**Keywords:** hyperuricemia, hypertension, complications

## Abstract

Hypertension remains a global threat to public health, affecting the worldwide population. It is one of the most common risk factors for cardiovascular disease. Today’s treatments focus on creating a hypotensive effect. However, there is a constant search for additional factors to reduce the potential of developing hypertension complications. These factors may act as a parallel treatment target with a beneficial effect in specific populations. Some studies suggest that uric acid may be considered such a factor. This study investigated the potential effect of uric acid concentrations over 5 mg/dL on the incidence of hypertension complications among patients with very high cardiovascular risk. A total of 705 patients with hypertension and very high cardiovascular risk were selected and included in the analysis. The patients were divided and compared according to serum uric acid levels. The study showed a higher occurrence of heart failure (OR = 1.7898; CI: 1.2738–2.5147; *p* = 0.0008), atrial fibrillation (OR = 3.4452; CI: 1.5414–7.7002; *p* = 0.0026) and chronic kidney disease (OR = 2.4470; CI: 1.3746–4.3558; *p* = 0.0024) among individuals with serum uric acid levels over 5 mg/dL, males and those with a BMI > 25 kg/m^2^. These findings suggest that even serum uric acid concentrations over 5 mg/dL may affect the prevalence of hypertension-related complications among patients with very high cardiovascular risk.

## 1. Introduction

Hypertension is a global problem affecting nearly 1.13 billion people worldwide, and it is estimated that by the year 2025, this number will reach over 1.56 billion. The World Health Organization data show that it is linked with 13% of deaths globally [1].

Even though prevention and treatment are continuously developing, hypertension remains the primary risk factor for cardiovascular diseases. Its complications that lead to death are still significant public health and clinical challenges [2].

Furthermore, this condition is a significant cause of premature death worldwide, mainly due to coronary artery disease (CAD) development. Nowadays, focusing on prevention and antihypertensive treatment is essential for dealing with hypertension. These actions were recently enhanced by tightening the target blood pressure (BP) for hypertensive patients and by intensifying the pharmacological treatments using initial one-pill combined therapies [3]. However, simultaneously with the evolution of the treatment of hypertension, scientists have been searching for additional factors that may decrease the probability of developing complications [2,3,4].

Uric acid (UA), which is the end-product of purine mononucleotide metabolism, has been the target of studies for many years. Purines can be synthesized endogenously or delivered from dietary sources. UA homeostasis is regulated by the balance between the rate of UA generation (determined by purine catabolism), renal excretion, and intestinal secretion. Balance disruption leads to an increase in serum uric acid (sUA) concentration, defined as hyperuricemia, which may consequently result in gout [5,6].

Many studies suggest that a higher concentration of sUA decreases hypertension treatment effectiveness and decreases cardiovascular complications and total number of deaths. However, despite several studies, the treatment of hyperuricemia remains unclear and inaccurate. Whether the treatment of asymptomatic hyperuricemia is cost-effective and whether the benefits of it prevail over the risks are still a matter of debate. A similar problem applies to the cut-off point for starting the treatment, the choice of the drug, its dosage, and the target value [2,7,8].

The cut-off value for hyperuricemia recognition is 6 mg/dL in men and 7 mg/dL in women. However, some sources show that lower values may be beneficial for a specific group of patients [9]. Asymptomatic hyperuricemia seems to be especially harmful to patients with high or very high cardiovascular risk [10]. In 2017 in Poland, experts initiated the idea of treating asymptomatic hyperuricemia in this group of patients, suggesting to start the treatment at a value of 5 mg/dL [8]. However, the evidence for such an approach is not sufficient. Therefore, further investigation of this matter is crucial.

The aim of our study was to assess the effect of hyperuricemia (UA > 5 mg/dL) on the prevalence of hypertension complications among patients with very high cardiovascular risk.

## 2. Materials and Methods

### 2.1. Study Design

The analysis was conducted from 27 December 2007 to 30 May 2016 as a case–control study. Data were collected by the Invasive Cardiology Department of the Medical University of Bialystok. The sample size consisted of 3291 patients with chronic coronary disease (CCD) with a Canadian Cardiovascular Society (CCS) Angina Score from I to III, and from this group, 705 patients were selected. The calculated sample size with a 5% margin of error and confidence level of 95% was equal to 316 (Sample Size Calculator—calculator.net). Inclusion criteria were admission for further invasive diagnostic process or invasive treatment, hypertension and echocardiography performed during hospitalization. Exclusion criteria were acute myocardial infarction, unstable angina, tumor lysis syndrome or in induction phase of chemotherapy. Our study population was divided into two subgroups—(I) uric acid in serum > 5 mg/dL and (II) uric acid in serum ≤ 5 mg/dL—according to the recommendations of Polish experts on treating asymptomatic hyperuricemia [8]. Laboratory tests, echocardiography, coronarography, and past medical history were included in the analysis. The study focused on the comparison of hypertension complications between the groups. Very high risk was defined according to cardiovascular disease prevention guidelines of the European Society of Cardiology due to the presence of atherosclerotic cardiovascular disease in subjects of the study [11].

### 2.2. Statistical Analysis

Continuous variables are expressed as median ± standard deviation. Categorical variables are expressed as percentages (number of patients). All parameters were tested for a normal distribution using the Kolmogorov–Smirnov test. To compare parametric continuous variables, we used the student’s *t*-test. The Mann–Whitney U-test was performed to compare nonparametric continuous variables and to compare categorical variables; the Odds Ratio Altman calculation was used. The statistical analysis was performed using Statistica 10. *p*-value ≤ 0.05 was considered significant.

## 3. Results

The median age of the patients was equal to 67 ± 9.85 y.o. Males constituted 65.81% of the study population. Our group was characterized by typical coronary artery disease comorbidities (Table 1). Over 80% of our study group had a BMI > 25 kg/m^2^, and over 65% had lipid disorders. Moreover, a history of myocardial infarction and heart failure occurred in almost half of the population (46.38% and 43.40%, respectively). Other frequent complaints were diabetes mellitus (35.03%), chronic kidney disease (13.82%), atrial fibrillation (8.83%), and chronic obstructive pulmonary disease (4.26%).

Most of our study population (71.51%) had serum uric acid concentrations above 5 mg/dL. In 192 cases (27.35%), the level was over 7 mg/dL, and only 18 patients (2.56%) demonstrated high, over 10 mg/dL, levels in the serum. In our study, we noticed that male gender (53.20% vs. 70.52%; *p* < 0.001), BMI > 25 kg/m^2^ (73.89% vs. 82.07%; *p* = 0.02), chronic kidney disease (7.39% vs. 16.33%; *p* = 0.002) and chronic obstructive pulmonary disease (2.96% vs. 6.77%; *p* = 0.054) were more frequently connected with uric acid > 5 mg/dL (Table 2). However, there was no difference in the stages according to KDIGO (Table 2).

While comparing hypertension complications between our two groups, we observed that heart failure (33.50% vs. 47.41%; *p* < 0.001) and atrial fibrillation (10.96% vs. 3.45%; *p* = 0.003) occurred more often in patients with uric acid in serum > 5 mg/dL (Table 3). However, the differences between those two groups regarding the prevalence of left ventricular hypertrophy (43.35% vs. 45.82%; *p* = 0.551), myocardial infarction history (41.38% vs. 48.41%; *p* = 0.09), carotid artery stenosis (12.81% vs. 12.35%; *p* = 0.868), limb ischemia (6.40% vs. 8.57%; *p* = 0.338) and stroke history (6.40% vs. 6.18%; *p* = 0.91) were not statistically significant (Table 3). Moreover, we investigated ischemia and no obstructive coronary artery disease (INOCA) in both groups with the same CAD risk factors. Our study shows that the group with higher uric acid levels in serum are predisposed to obstructive coronary artery disease rather than no obstructive coronary artery disease (INOCA: 35.47% vs. 27.09%; *p* = 0.028), without an effect on the number of affected vessels (Table 3).

Usage of diuretics and beta-blockers in patients with uric acid concentrations > 5 mg/dL was higher than in those with a lower level of uric acid in their serum (83.25% vs. 92.43%; *p* < 0.001 and 35.96% vs. 62.95%; *p* < 0.001, respectively) (Table 4). Therefore, patients with uric acid concentrations > 5 mg/dL needed more medication to reach the same hypertension treatment goal than patients with uric acid levels ≤ 5 mg/dL.

However, we found no difference in age (67.00 ± 9.26 y.o. vs. 67.00 ± 10.09 y.o., *p* = 0.947), systolic blood pressure (134.00 ± 22.83 mmHg vs. 134.00 ± 21.36 mmHg, *p* = 0.875), diastolic blood pressure (76.50 ± 13.9 mmHg vs. 78.00 ± 13.10 mmHg, *p* = 0.128) and blood glucose level (107.50 ± 51.91 mg/dL vs. 104.00 ± 46.68 mg/dL, *p* = 0.267) between subject with sUA concentrations ≤ 5 and >5. Furthermore, the patients with sUA levels over 5 had higher heart rates and more atherogenic lipid profiles (higher LDL and triglycerides, lower HDL).

Echocardiography showed that patients with uric acid levels > 5 mg/dL had a lower left ventricular ejection fraction (55.00 ± 9.14% vs. 50.00 ± 11.26%; *p* < 0.001), and larger diameters for the left atrium (38.5 ± 5.96mm vs. 40 ± 7.21mm; *p* = 0.003), ascending aorta and both left and right ventricles (Table 5). Moreover, a trend towards a higher frequency of eccentric hypertrophy occurrence (19.21% vs. 25.90%; *p* = 0.061) was noticeable (Table 4).

## 4. Discussion

In our study, we assessed the influence of serum uric acid levels in a group of patients with a very high cardiovascular risk. Numerous studies have shown that hyperuricemia is associated with CVD such as hypertension, coronary artery disease, peripheral vascular disease, heart failure, and stroke [12,13]. However, some studies are suggesting that the effect of hyperuricemia is related to comorbidity with other diseases [14].

Uric acid is included in the Guidelines of the European Society of Cardiology as one of the hypertension risk factors [2]. Our study results show that quantitatively, the majority of the population with a very high cardiovascular risk and coexisting hypertension have increased uric acid levels above 5 mg/dL. Our results may suggest that even a small increase in the concentration of uric acid can be a significant risk factor for the development of hypertension and its complications in a high-risk population.

Two main factors that impact the increase in uric acid levels among the hypertension population are male gender and BMI above 25 kg/m^2^ (Table 2). The association between the male gender and hyperuricemia is well known and was confirmed by other authors. Gender is an unmodifiable risk factor; therefore, in males with very high cardiovascular risk and coexisting hypertension, serum uric acid levels testing should be performed routinely for early detection or effective prophylaxis [10]. In the case of patients with a variety of comorbidities with high cardiovascular risk, maintaining a proper BMI level is crucial to protect from hyperuricemia and its complications. The positive association between BMI and sUA levels was confirmed by Nurshad Ali et al. as well as Chizyński et al. [15,16]. However, according to Ryuichi Kawamoto et al.’s research, a higher BMI is not associated with hypertension in the hyperuricemic population [17].

The subject with sUA levels over 5 mg/dL had a higher heart rate (HR). It is an especially important implication because the HR over 80 beats/min was included in factors influencing cardiovascular risk in patients with hypertension by the European Society of Cardiology and the European Society of Hypertension [2]. Also, the Working Group on Uric Acid and Cardiovascular Risk of the Italian Society of Hypertension performed Multivariable Cox analyses on 19,128 patients with higher heart rates and it was associated with a higher risk in patients with hyperuricemia, independent of hypertension, age, and taking beta-blockers [18]. In subjects with HR ≥76 beats/min, the relative risk was equal to 2.38 (95% CI 1.82–3.10) [18].

In our study, patients with elevated sUA concentrations had an increased prevalence of atrial fibrillation (AF). Most atrial fibrillation cases occurred in the group with sUA levels > 5 mg/dL (88.70%). Moreover, the sUA level was significantly associated with AF in the general Japanese population [19]. Hyperuricemia was found to increase the risk of heart rate irregularity in the Korean National Health and Nutrition Examination Survey [20]. Our study shows that an elevated concentration of sUA intensifies this effect. We further confirmed this association between increased sUA levels and AF, independent of age. Although this mechanism is not well understood, Yue Chen et al. suggest that UA is associated with processes that may have an impact on cardiac geometry—especially the left atrium [21]. The results of our research also show this relationship between an increased concentration of sUA and cardiac remodeling—eccentric left ventricular hypertrophy and left atrium dimension enlargement.

Analyzing the results, we discovered that patients with sUA concentrations above 5 mg/dL are significantly more likely to have chronic kidney disease. The relationship between asymptomatic hyperuricemia and CKD is confirmed by other researchers [22,23]. Moreover, an increased sUA concentration is responsible not only for the development but also for the progression of chronic kidney disease [24,25]. In a meta-analysis based on 11 papers with a total of only 753 participants, Wang et al. reported that a decrease in sUA concentration is associated with a significant lowering of the serum creatinine concentration and an increase in the eGFR [26]. This leads to the conclusion that effective treatment and prevention of hyperuricemia (from 5 mg/dL in patients with increased cardiovascular risk) protects against disease and further progression.

Moreover, our study shows that uric acid level was not related to the development of limb ischemia, carotid artery stenosis, or stroke. Numerous reports have shown that peripheral arterial disease (PAD) is found more often in patients with a higher concentration of uric acid in the serum [27,28]. Langlois et al. suggest that an increased sUA level is a complication of PAD [29]. According to other study, PAD factors are mainly diabetes mellitus and smoking [30]. Our study indicates that more prospective studies must be conducted to investigate this matter.

As mentioned earlier, sUA concentration is a risk factor for cardiovascular diseases, including hypertension [2,7,8]. So far, there have been few studies performed on the influence of uric acid concentration on the occurrence of heart failure. Our study shows that higher uric acid levels in combination with hypertension may lead to an increased risk of heart failure with reduced ejection fraction presence. In combination with hypertension, uric acid concentration has a significant impact on the occurrence of heart failure with reduced ejection fraction. Borghi et al. additionally present their theory from a pathophysiological point of view. They claim that sUA functionally up-regulates xanthine oxidase (XO), which is a key enzyme in purine metabolism. XO-derived reactive oxygen species may account for the range of detrimental processes in the development of heart failure, such as cardiac hypertrophy, myocardial fibrosis, left ventricular remodeling and contractility impairment [31]. Our study has shown no association between sUA concentration > 5 mg/dL and multivessel disease. There is no common consensus on the influence of UA on the prevalence of coronary artery disease. The study conducted by De Luca et al. revealed that sUA levels do not affect the prevalence and extent of coronary artery disease [32]. On the other hand, Tian et al. proved in a prospective study that a high sUA level is an independent risk factor for the presence and severity of early-onset coronary artery disease [33]. We noticed that patients with elevated sUA concentrations were more likely to be characterized by obstructive coronary artery disease. We also searched for a link between multi-vascular coronary artery disease and elevated sUA levels. Lan et al. in their research proved that hyperuricemia has an impact on multivessel disease only in women [34].

The relationship between lipid profile and serum uric acid remains unclear; a similar study concerning patients with very high risk found no correlation between uric acid and triglycerides [10]. However, in the sub-analysis of the data collected in the NHANES III study, after adjusting, the LDL, triglyceride and total cholesterol levels were higher in individuals with higher sUA concentrations, similar to our study [35]. Despite that, in our study, both groups had high LDL cholesterol levels for patients with very high risk, extending the treatment goal. The relationship between uric acid and dyslipidemia is especially interesting because the long uninterrupted treatment with statins was found to decrease the risk of developing gout [36]. According to our study, patients with hypertension and increased levels of sUA used more drugs, especially beta-blockers and diuretics. Ueno et al. in their work have proven that these drugs increased the level of sUA by reducing the glomerular filtration rate, which may have contributed to further complications. They claim that ACE-I and sartans do not cause this undesirable effect. Therefore, they suggest using ACE-I and sartans as primary hypotensive agents [37]. However, in the presence of significant indications for the use of beta-blockers and diuretics, drugs that directly reduce uric acid levels can be included in the treatment. Such an approach may be supported by a study showing positive effects and an increased survival rate [38]. Larsen et al. reported that allopurinol treatment is associated with decreased cardiovascular risk among hyperuricemic patients [39]. Li Zhao et al. also supported that statement [14]. However, a recent study, ALL-HEART, revealed no differences in primary outcomes of cardiovascular disease-related mortality between groups randomized to allopurinol treatment and usual care [40]. These ambiguous results lead us to believe that there is still a special need for studying the benefits of intensive and early uric acid-lowering therapy, especially in high-risk patients.

A novel approach of using Sodium-Glucose Cotransporter 2 (SGLT-2) inhibitors to lower uric acid was suggested, and this effect was proposed as one of the possible components of SGLT-2 inhibitors’ pleiotropic effects on cardiovascular diseases by anti-inflammatory actions [41]. The SGLT-2 inhibitors alone were found to significantly decrease sUA levels in patients with diabetes mellites [42]. Furthermore, combining the SGLT-2 inhibitors with sUA-lowering medications showed an additive effect [42]. The double-blind prospective studies involving the effect of SGLT-2 inhibitors on cardiovascular risk in patients with hyperuricemia alone or combined with hypertension should be performed.

## 5. Conclusions

Our study shows that even a slight increase in uric acid concentration (even over 5 mg/dL) in patients with hypertension and very high cardiovascular risk further increases the risk of organ complications. Therefore, early hyperuricemia treatment and sUA level monitoring to prevent multiple organ complications among those patients should be considered. Large prospective, double-blind studies designed to assess the effect of decreasing sUA early among hypertensive patients with very high cardiovascular risk should be conducted.

## 6. Study Limitations

### 6.1. Study Design

The study was performed in a single center, the subjects originate from Eastern Europe and all participants were Caucasian; thus, the results cannot be generalized to other ethnicities.

The study analyzed the difference between groups with sUA concentrations ≤ 5 mg/dL and >5 mg/dL, and the possible effect of hyperuricemia treatment on hypertensive patients must be verified by prospective randomized trials.

An additional limitation of our study is that the analysis did not take sex into account.

### 6.2. Data Collection

The retrospective aspect of the study, based on data acquired from medical documentation does not allow for including the whole initial population due to lack of data regarding sUA concentrations.

## Figures and Tables

**Table 1 healthcare-11-02460-t001:** Basic study population characteristics.

Basic Population Characteristics			
No. of patients	705	Age	67.00 ± 9.85 years
Male	65.81% (462)	Atrial fibrillation	8.83% (62)
Chronic obstructive pulmonary disease	4.26% (30)	Heart failure	43.40% (306)
BMI > 25 kg/m^2^	80.06% (562)	Past myocardial infarction	46.38% (327)
Chronic kidney disease	13.82% (97)	Uric acid in serum >5 mg/dL	71.51% (502)
Diabetes mellitus	35.03% (247)	Uric acid in serum >7 mg/dL	27.35% (192)
Lipid and cholesterol disorders	67.23% (474)	Uric acid in serum >10 mg/dL	2.56% (18)

**Table 2 healthcare-11-02460-t002:** General characteristics of the groups.

General Characteristic	UA ≤ 5 mg/dL *n* = 203	UA > 5 mg/dL *n* = 502	OR	95% Cl	*p*-Value
BMI > 25 kg/m^2^	73.89% (150)	82.07% (412)	1.5823	1.0754 to 2.3283	0.02
Male	53.20% (108)	70.52% (354)	2.1040	1.5037 to 2.9438	<0.001
Gout	0.49% (1)	3.19% (16)	6.6502	0.8761 to 50.4822	0.067
Hyperuremic treatment	4.93% (10)	5.58% (28)	1.1401	0.5433 to 2.3924	0.729
Diabetes mellitus	32.51% (66)	36.06% (181)	1.1704	0.8284 to 1.6536	0.372
Lipid and cholesterol disorder	70.94% (144)	65.74% (330)	0.7861	0.5514 to 1.1207	0.184
Asthma	2.96% (6)	2.39% (12)	0.8041	0.2976 to 2.1723	0.667
Chronic obstructive pulmonary disease	2.96% (6)	6.77% (34)	2.3853	0.9857 to 5.7722	0.054
Sleep apnea	0.00% (0)	1.00% (5)	4.4995	0.2477 to 81.7502	0.309
Hyperthyroidism	4.43% (9)	3.78% (19)	0.8479	0.3771 to 1.9069	0.690
Hypothyroidism	2.96% (6)	1.99% (10)	0.7345	0.2633 to 2.0491	0.556

**Table 3 healthcare-11-02460-t003:** Hypertension-related complication comparison.

Hypertension-Related Complication	UA ≤ 5 mg/dL *n* = 203	UA > 5 mg/dL *n* = 502	OR	95% Cl	*p*-Value
Left ventricular hypertrophy:					
Total	43.35% (88)	45.82% (230)	1.1050	0.7957 to 1.5347	0.551
Concentric	24.14% (49)	19.92% (100)	0.7818	0.5299 to 1.1536	0.215
Eccentric	19.21% (39)	25.90% (130)	1.4695	0.9828 to 2.1972	0.061
Peripheral artery disease (PAD):					
Total	19.21% (39)	20.92% (105)	1.1122	0.7380 to 1.6760	0.611
Carotid artery stenosis	12.81% (26)	12.35% (62)	0.9593	0.5876 to 1.5659	0.868
Limb ischemia	6.40% (13)	8.57% (43)	1.3692	0.7198 to 2.6044	0.338
Renal:					
Chronic kidney disease	7.39% (15)	16.33% (82)	2.4470	1.3746 to 4.3558	0.002
G1	0% (0)	0% (0)	0.1879	0.0036 to 9.8298	0.4076
G2	0% (0)	8.54% (7)	3.0795	0.1670 to 56.7825	0.4494
G3A	53.33% (8)	34.15% (28)	0.4537	0.1492 to 1.3799	0.1638
G3B	20% (3)	37.80% (31)	2.2288	0.5764 to 8.6180	0.2454
G4	20% (3)	17.07% (14)	0.7549	0.1861 to 3.0620	0.6939
G5	6.67% (1)	2.44% (2)	0.1728	0.0102 to 2.9268	0.2240
Albuminuria	1.97% (4)	1.99% (10)	1.0112	0.3135 to 3.2619	0.985
Myocardial infarction history	41.38% (84)	48.41% (243)	1.3292	0.9559 to 1.8481	0.091
INOCA	35.47% (72)	27.09% (136)	0.6761	0.4772 to 0.9579	0.028
Stroke history	6.40% (13)	6.18% (31)	0.9619	0.4926 to 1.8783	0.91
Heart failure:					
Total	33.50% (68)	47.41% (238)	1.7898	1.2738 to 2.5147	<0.001
Heart failure with preserved ejection fraction (HFpEF)	14.29% (29)	12.75% (64)	0.8767	0.5465 to 1.4064	0.878
Heart failure with mildly reduced ejection fraction (HFmEF)	8.87% (18)	16.14% (81)	1.9774	1.1533 to 3.3906	0.013
Heart failure with reduced ejection fraction (HFrEF)	9.85% (20)	17.33% (87)	1.9182	1.1447 to 3.2143	0.013
Atrial fibrillation	3.45% (7)	10.96% (55)	3.4452	1.5414 to 7.7002	0.003

**Table 4 healthcare-11-02460-t004:** Hypertension treatment.

Hypertension Treatment	UA ≤ 5 mg/dL *n* = 203	UA > 5 mg/dL *n* = 502	OR	95% Cl	*p* Value
ACE-I	73.89% (150)	77.89% (391)	1.2446	0.8533 to 1.8155	0.256
ARB	17.73% (36)	18.92% (95)	1.0828	0.7088 to 1.6542	0.713
Ca blockers	47.78% (97)	42.83% (215)	0.8186	0.5902 to 1.1356	0.231
Beta blockers	83.25% (169)	92.43% (464)	2.4566	1.4973 to 4.0303	<0.001
Diuretics	35.96% (73)	62.95% (316)	3.0255	2.1554 to 4.2468	<0.001

**Table 5 healthcare-11-02460-t005:** Baseline characteristics.

Variable	UA ≤ 5 mg/dL *n* = 203	UA > 5 mg/dL *n* = 502	*p*-Value
Systolic BP (mm Hg)	134.00 ± 22.83	134.00 ± 21.36	0.875
Diastolic BP (mm Hg)	76.50 ± 13.9	78.00 ± 13.10	0.128
Pulse Pressure (mm Hg)	57.00 ± 19.32	56.00 ± 18.61	0.579
Mean Arterial Pressure (mm Hg)	95.16 ± 13.65	97.00 ± 13.3	0.445
Heart Rate (beats/minute)	64.00 ± 9.33	66.00 ± 13.60	0.015
Serum Uric Acid (mg/dL)	4.26 ± 0.65	6.58 ± 1.43	<0.001
Age (years)	67.00 ± 9.26	67.00 ± 10.09	0.947
BMI (kg/m^2^)	27.59 ± 4.29	29.02 ± 4.43	<0.001
Creatinine (mg/dL)	0.82 ± 0.74	0.98 ± 0.49	<0.001
eGFR (ml/min/m^2^)	83.00 ± 28.06	72.00 ± 32.37	0.006
Blood Glucose (mg/dL)	107.50 ± 51.91	104.00 ± 46.68	0.268
Total Cholesterol (mg/dL)	158.5 ± 42.60	161.00 ± 41.88	0.2794
LDL (mg/dL)	85.00 ± 36.38	93.00 ± 36.17	0.034
HDL (mg/dL)	49.50 ± 13.29	44.00 ± 22.25	<0.001
Triglycerides (mg/dL)	107.00 ± 55.93	161.00 ± 90.62	<0.001
LVEF (%)	55.00 ± 9.14	50.00 ± 11.26	<0.001
LA (mm)	38.5 ± 5.96	40.00 ± 7.21	0.003
RVDD (mm)	29.00 ± 3.46	30.00± 3.53	<0.001
Ascending Aorta (mm)	34.00 ± 3.86	35.00 ± 4.04	0.003
LVEDD (mm)	48.00 ± 6.23	50.00 ± 6.77	<0.001
Intraventricular Septum (mm)	11.00 ± 1.54	11.00 ± 1.65	0.6340
Posterior Wall (mm)	11.00 ± 1.19	11.00 ± 1.30	0.3881
LV Mass Index (g/m^2^)	100.00 ± 31.02	108.00 ± 29.16	0.0200

## Data Availability

The data used to support the findings of this study are available from the corresponding author.

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
