# Peer review of "Hyperuricemia as a Risk Factor in Hypertension among Patients with Very High Cardiovascular Risk"

_healthcare, 2023, doi:10.3390/healthcare11172460_

Round 1
Reviewer 1 Report
1. Language revision is highly recommended.
2. Line no: 42, "However, simultaneously with the evolution ........ but also may increase the probability of developing its complications" needs to be reframed since it is seems to be misleading.
3. The novelty and significance of the current study needs more justification and validation, since the association of uric acid concentration and cardiovascular complications are well established.
4. The authors are suggested to use serum instead of blood serum throughout the manuscript.
5. The study included patients with high cardiovascular risk. How do authors claim all the complications to be associated with hypertension alone, what about other factors? There are several other factors like obesity that contributes to cardiovascular diseases.
6. There was no significant difference in blood pressure between the groups. How do the authors correlate hypertension and uric acid levels?
The manuscript needs language revision.
Author Response
1. Language revision is highly recommended.
Reply 1: The manuscript has been thoroughly revised.
2. Line no: 42, "However, simultaneously with the evolution ........ but also may increase the probability of developing its complications" needs to be reframed since it is seems to be misleading.
Reply 2: Thank you for this comment. This sentence was simplified not to mislead potential readers.
3. The novelty and significance of the current study needs more justification and validation, since the association of uric acid concentration and cardiovascular complications are well established.
Reply 3: We understand the reviewer's concerns. However, we believe that our study presents significant novelty. First of all, our study is unique in that it analyzes the effects of different concentrations of uric acid. Secondly, there are conflicting results about the influence of uric acid lowering therapy on the prognosis in the available literature. Most of the studies suggest that higher concentrations of uric acid are associated with worse survival. Nevertheless, the most recent randomized clinical trial conducted by Mackenzie et al. proved that conventional therapy with allopurinol did not improve the prognosis of patients with ischemic heart disease without concomitant gout. These points suggest that the treatment used so far is ineffective and other groups of drugs should be sought. In the discussion, we have added interesting consideration about the use of SGLT-2 inhibitors in the treatment of hyperuricemia, as it was one of their suggested pleiotropic effects. In light of all of these points, we believe that there is still a need for observational studies to further cement the negative effects of hyperuricemia, as well as randomized control trials to find new ways of treatment.
4. The authors are suggested to use serum instead of blood serum throughout the manuscript.
Reply 4: Thank you for the suggestion, we corrected the terminology used throughout the manuscript.
5. The study included patients with high cardiovascular risk. How do authors claim all the complications to be associated with hypertension alone, what about other factors? There are several other factors like obesity that contributes to cardiovascular diseases.
Reply 5: Thank you for the important question. Our study included patients with high cardiovascular risk, the aspect of comorbidities’ etiology is multi-dimensional and we agree that not all of the complications are associated with hypertension alone. In order to clarify this issue we have now updated terminology to ‘hypertension-related complications’.
6. There was no significant difference in blood pressure between the groups. How do the authors correlate hypertension and uric acid levels?
Reply 6: Thank you for the comment. All of the patients included in the study had a history of hypertension and were treated according to the pertinent guidelines, therefore no significant difference in blood pressure may be related with the pharmacological treatment. Hyperuricemia is an established modifiable cardiovascular risk factor. The purpose of our study was to assess the effect of hyperuricemia (UA>5 mg/dl) on the prevalence of hypertension complications among patients with very high cardiovascular risk, not the blood pressure levels itself.
Reviewer 2 Report
Dear authors,
The manuscript presented to me for evaluation is very interesting and meets the thematic requirements of the journal. However, the authors made a few minor errors related to the manuscript's in adjustment to the journal requirements, among others, they did not explain the abbreviations used for the first time in the text of the manuscript (e.g. in the abstract). The same applies to tables.
In lines 109-112, it is worth emphasizing that the comparison concerns patients with different levels of UA.
The list of references and citations should be adapted to the requirements of the journal, among others line 194 no references assigned, names not cited correctly, some authors omitted from references.
In addition, it is worth expanding the aspect of laboratory tests, echocardiography, coronary angiography and describing the history of the disease in more detail.
In my opinion, there is too much data from Table 3 cited in the the manuscript - this is a repetition (if possible please remove it).
It is worth complementing the work with the strengths and weaknesses of the manuscript.
After applying appropriate corrections (according to the journal's recommendations), the paper should be published.
Wish you all the best.
Sincerely,
Reviewer
Author Response
Dear authors,
The manuscript presented to me for evaluation is very interesting and meets the thematic requirements of the journal. However, the authors made a few minor errors related to the manuscript's in adjustment to the journal requirements, among others, they did not explain the abbreviations used for the first time in the text of the manuscript (e.g. in the abstract). The same applies to tables.
Reply: Thank you for the time and effort invested in reviewing our manuscript
Comment 1: In lines 109-112, it is worth emphasizing that the comparison concerns patients with different levels of UA.
Reply 1: Thank you for the suggestion, please see the manuscript amended.
Comment 2: The list of references and citations should be adapted to the requirements of the journal, among others line 194 no references assigned, names not cited correctly, some authors omitted from references.
Reply 2: Thank you, we apologize for the omission. Now we have updated the references accordingly.
Comment 3: In addition, it is worth expanding the aspect of laboratory tests, echocardiography, coronary angiography and describing the history of the disease in more detail.
Reply 3: Thank you for this suggestion. We expanded the aspect of laboratory and echocardiographic characteristics. However, we decided not to expand coronary angiography section due to the purpose of the study – assessment of the effect of hyperuricemia on the prevalence of hypertension complications.
Comment 4: In my opinion, there is too much data from Table 3 cited in the the manuscript - this is a repetition (if possible please remove it).
Reply 4: We understand your concerns. However, we aimed to present the results as accurately as possible. Only the most important results that we wanted to highlight are repeated. Moreover, the strict conditions of the journal regarding the required number of words do not allow us to shorten the manuscript.
Comment 5: It is worth complementing the work with the strengths and weaknesses of the manuscript.
Reply 5: Thank you for this suggestion. We have implemented limitations of our study.
Reviewer 3 Report
Comments to the Author
The authors performed a case-control study and investigated the potential effect of uric acid over 5 mg/dl on the incidence of hypertension complications among patients with very high cardiovascular risk. The reviewer has some comments/concerns and recommended major revisions.
1. It is noticeable that gender is not matching in the two groups (UA< 5 and UA>5) and we know gender is an important factor for uric acid. Did the author include gender as a covariate in statistical analysis? How did the authors address the effect of gender?
2. Many health markers have been compared between the UA< 5 mg/dl and UA>5 mg/dl group, however, it is not clear that which one is the primary outcome, and which are secondary outcomes.
3. This is an unmatched case-control study. How did the author estimate the sample size? It is highly recommended to include the sample size estimation method in the Materials and Methods section. In addition, many outcomes have been evaluated in this manuscript, the authors need verify the current sample size could provide sufficient power for testing so many outcomes.
4. Table 1: Since the most comparisons were performed between two groups (UA<5 or UA>5, it is highly recommended to present the basic population characteristics in two groups. It seems Tables 1, 2 and % are all about baseline characteristics, which may be combined to one Table. What is COPD? All abbreviations in the tables should be annotated below the table. It is recommended to add ethnicity and race information.
n/a
Author Response
Comments to the Author
The authors performed a case-control study and investigated the potential effect of uric acid over 5 mg/dl on the incidence of hypertension complications among patients with very high cardiovascular risk. The reviewer has some comments/concerns and recommended major revisions.
- It is noticeable that gender is not matching in the two groups (UA< 5 and UA>5) and we know gender is an important factor for uric acid. Did the author include gender as a covariate in statistical analysis? How did the authors address the effect of gender?
Reply 1: Thank you for the comment. We agree with the Reviewer that gender is an important unmodifiable cardiovascular risk factor and is also associated with the uric acid concentrations. The goal of our study was to assess the effect of hyperuricemia (UA>5 mg/dl) on the prevalence of hypertension complications among the overall cohort of patients with very high cardiovascular risk. We agree that lack of adjustment for sex may be a limitation of our study and we have now updated limitations section of our manuscript.
- Many health markers have been compared between the UA< 5 mg/dl and UA>5 mg/dl group, however, it is not clear that which one is the primary outcome, and which are secondary outcomes.
Reply 2: Thank you for this comment. We strongly agree that the association between uric acid concentration and co-morbidities is a complex issue. Our study is a retrospective analysis of patients with very high cardiovascular risk. The reviewer’s question is very important, however, due to the retrospective design we cannot draw such conclusions from our study. In order to clarify this issue large prospective, double-blind studies should be conducted.
- This is an unmatched case-control study. How did the author estimate the sample size? It is highly recommended to include the sample size estimation method in the Materials and Methods section. In addition, many outcomes have been evaluated in this manuscript, the authors need verify the current sample size could provide sufficient power for testing so many outcomes.
Reply 3: Thank you for your suggestion. We have significantly improved the Materials and Methods section, please notice changes in the manuscript.
- Table 1: Since the most comparisons were performed between two groups (UA<5 or UA>5, it is highly recommended to present the basic population characteristics in two groups. It seems Tables 1, 2 and % are all about baseline characteristics, which may be combined to one Table. What is COPD? All abbreviations in the tables should be annotated below the table. It is recommended to add ethnicity and race information.
Reply 4: We understand your concerns. However, we have avoided merging this data into one table not to mislead potential readers. On the other hand, all of the abbreviations are annotated and we have added ethnicity and race information.
Reviewer 4 Report
2.1 Study design. Inclusion criteria- CCD could be deleted, as the initial sample size comprised patients with CCD.
Material and methods – as a title of the topics deals with the arterial hypertension authors need to give more datails what stages of AH patients had and what class of CCD patients had
Table 1. If add all 3 groups with different levels of uric acid, the total number of the patients would be 712, not 705. Recheck
It is not needed to divide patients into groups with UA level >7 mg/dl and >10
mg/dl but to have only one group with UA level more than 5 mg. As the patients with the UA level >7 mg/dl and >10 mg/dl are not analyzed in paper.
The text after the table 2 - “However, we found no difference in age………” These data belong to table 5. I suggest to put the table 5 after the table 2 as table 3.
Table 1 and 2. Lipid and cholesterol disorder. It is better to give separately total Col., LDL-C, HDL-C, TG
Table 3 Hypertension complications comparison. What stages of CKD were presented? What was eGFR? No explanation of abbreviations HFpEF, HFmEF, HFrEF below the table.
Ref. 6. In this reference the names of the authors were given in capital letters
Ref. 19. In this reference the year of the journal is given at the end but in other references the years are given just after the title of the journal.
Author Response
2.1 Study design. Inclusion criteria- CCD could be deleted, as the initial sample size comprised patients with CCD.
Reply: Thank you, we have followed the suggestions.
Material and methods – as a title of the topics deals with the arterial hypertension authors need to give more datails what stages of AH patients had and what class of CCD patients had
Reply: We agree with the reviewer's suggestion. However, we were unable to assess the initial values of blood pressure before the introduction of hypertension treatment, therefore we cannot classify them into appropriate groups. The vast majority of recruited patients were people with already implemented antihypertensive treatment.
Table 1. If add all 3 groups with different levels of uric acid, the total number of the patients would be 712, not 705. Recheck
Reply: Patients may be represented more than once since patients with UA level >7 mg/dl and >10 mg/dl also qualifies for the >5 mg/dl group.
It is not needed to divide patients into groups with UA level >7 mg/dl and >10
mg/dl but to have only one group with UA level more than 5 mg. As the patients with the UA level >7 mg/dl and >10 mg/dl are not analyzed in paper.
Reply: We understand the reviewer’s concerns. However, our goal in including these subgroups was to accurately present the analyzed population.
The text after the table 2 - “However, we found no difference in age………” These data belong to table 5. I suggest to put the table 5 after the table 2 as table 3.
Reply: We have followed the suggestions, please notice suggestions to our manuscript.
Table 1 and 2. Lipid and cholesterol disorder. It is better to give separately total Col., LDL-C, HDL-C, TG
Reply: Thank you for this comment. Please see Table 1 and Table 2 amended. We have also updated discussion section according to the new data.
Table 3 Hypertension complications comparison. What stages of CKD were presented? What was eGFR? No explanation of abbreviations HFpEF, HFmEF, HFrEF below the table.
Reply: Thank you for this comment, we have updated the missing informations.
Ref. 6. In this reference the names of the authors were given in capital letters
Reply: Thank you. We have now changed the references section accordingly.
Ref. 19. In this reference the year of the journal is given at the end but in other references the years are given just after the title of the journal.
Reply: Thank you. We have now changed the references section accordingly.
Reviewer 5 Report
Dear Authors,
Like you mentioned even though a lot of research has been conducted regarding the correlation of the serum uric acid levels and cardiovascular risk so far there is no compelling data to say that we need to focus on decreasing the serum UA levels to decrease the complications of the HTN and cardiovascular disease and its complications. More prospective analysis is needed to tell the correlation and include it in the treatment guidelines.
You have presented the data well for your patient cohort and will be useful in guiding the future studies in this topic eventhough the findings are not unique or new
Author Response
Dear Authors,
Like you mentioned even though a lot of research has been conducted regarding the correlation of the serum uric acid levels and cardiovascular risk so far there is no compelling data to say that we need to focus on decreasing the serum UA levels to decrease the complications of the HTN and cardiovascular disease and its complications. More prospective analysis is needed to tell the correlation and include it in the treatment guidelines.
You have presented the data well for your patient cohort and will be useful in guiding the future studies in this topic eventhough the findings are not unique or new
Reply: We would like to thank the reviewer for the kind comments. We believe that our study presents some novelty. First of all, our study is unique in that it analyzes the effects of different concentrations of uric acid. Secondly, there are conflicting results about the influence of uric acid lowering therapy on the prognosis in the available literature. Most of the studies suggest that higher concentrations of uric acid are associated with worse survival. Nevertheless, the most recent randomized clinical trial conducted by Mackenzie et al. proved that conventional therapy with allopurinol did not improve the prognosis of patients with ischemic heart disease without concomitant gout. These points suggest that the treatment used so far is ineffective and other groups of drugs should be sought. In the discussion, we have added interesting consideration about the use of SGLT-2 inhibitors in the treatment of hyperuricemia, as it was one of their suggested pleiotropic effects. In light of all of these points, we believe that there is still a need for observational studies to further cement the negative effects of hyperuricemia, as well as randomized control trials to find new ways of treatment.
Round 2
Reviewer 3 Report
The authors addressed most of my concerns. The p-values in the Table 5 have some typos.
n/a
Author Response
Thank you for the time and effort invested in reviewing our manuscript.
As suggested, we improved Table 5.
Reviewer 4 Report
Authors have made the corrections in article
Author Response
Thank you for the time and effort invested in reviewing our manuscript